# Probiotic *Bifidobacterium longum* subsp. *longum* Protects against Cigarette Smoke-Induced Inflammation in Mice

**DOI:** 10.3390/ijms24010252

**Published:** 2022-12-23

**Authors:** Kurtis F. Budden, Shaan L. Gellatly, Annalicia Vaughan, Nadia Amorim, Jay C. Horvat, Nicole G. Hansbro, David L. A. Wood, Philip Hugenholtz, Paul G. Dennis, Peter A. B. Wark, Philip M. Hansbro

**Affiliations:** 1Priority Research Centre for Healthy Lungs and Immune Health Research Program, Hunter Medical Research Institute and The University of Newcastle, Callaghan, NSW 2308, Australia; 2Centre for Inflammation, School of Life Sciences, Faculty of Science, Centenary Institute and University of Technology Sydney, Sydney, NSW 2007, Australia; 3Australian Centre for Ecogenomics, School of Chemistry and Molecular Biosciences, The University of Queensland, Brisbane, QLD 4102, Australia; 4School of Earth and Environmental Sciences, The University of Queensland, Brisbane, QLD 4072, Australia

**Keywords:** probiotic, *Bifidobacterium*, cigarette smoke, COPD, inflammation, SCFA, acetate

## Abstract

*Bifidobacterium* are prominent gut commensals that produce the short-chain fatty acid (SCFA) acetate, and they are often used as probiotics. Connections between the gut and the lung, termed the gut–lung axis, are regulated by the microbiome. The gut–lung axis is increasingly implicated in cigarette smoke-induced diseases, and cigarette smoke exposure has been associated with depletion of *Bifidobacterium* species. In this study, we assessed the impact of acetate-producing *Bifidobacterium longum* subsp. *longum* (WT) and a mutant strain with an impaired acetate production capacity (MUT) on cigarette smoke-induced inflammation. The mice were treated with WT or MUT *B. longum* subsp. *longum* and exposed to cigarette smoke for 8 weeks before assessments of lung inflammation, lung tissue gene expression and cecal SCFAs were performed. Both strains of *B. longum* subsp. *longum* reduced lung inflammation, inflammatory cytokine expression and adhesion factor expression and alleviated cigarette smoke-induced depletion in caecum butyrate. Thus, the probiotic administration of *B. longum* subsp. *longum,* irrespective of its acetate-producing capacity, alleviated cigarette smoke-induced inflammation and the depletion of cecal butyrate levels.

## 1. Introduction

The genus *Bifidobacterium* is comprised of strictly anaerobic Gram-positive rods [1,2]. Whilst they are particularly prominent in the gut microbiome during early life, especially amongst breastfed infants, their role in human health remains important in adulthood, and they are widely used as probiotics [2,3,4]. *Bifidobacterium* are considered to be beneficial components of the gut microbiome, inducing a range of immunoregulatory responses in the host, and promoting the effective clearance of bacterial and viral infections in the gastrointestinal tract, while limiting excessive inflammation [1,2]. Although the beneficial effects of *Bifidobacterium* spp. are often associated with their production of the short-chain fatty acid (SCFA) acetate [5], they can also regulate immune responses through other metabolites or direct interactions with immune cells via antigen presentation to host pattern recognition receptors [6,7].

The connections between the gut microbiome and the lungs are being increasingly well recognized, and cigarette smoking has a strong influence on the microbiome and the gut–microbiome–lung axis [8,9]. Gut microbiome changes are implicated in smoking-associated lung diseases such as chronic obstructive pulmonary disease (COPD) [10] and lung cancer [11], as well as inflammatory bowel [12] and non-alcoholic fatty liver [13] diseases. The abundance of *Bifidobacterium* and SCFAs (including acetate) are reduced by cigarette smoke exposure in humans [4,14,15] and rats [16]. Additionally, cigarette smoke condensate impaired the growth, exopolysaccharide and acetate production of *Bifidobacterium animalis* in vitro [17].

*Bifidobacterium longum* subsp. *longum* is an acetate-producing species that is commonly used as a probiotic [18], and it has been previously associated with a lower incidence of asthma [19], demonstrating a likely involvement in the gut–lung axis. Additionally, the insertional mutagenesis of an ABC-type transporter gene in *B. longum* subsp. *longum* NCC2705 produced a strain with a severely impaired ability to produce acetate [5]. This enabled us to distinguish between the effects mediated by acetate and those mediated by other bacterial products to define the underlying mechanisms by which this bacterium interacts with host immunity.

Given the critical role of smoking in the development of chronic respiratory and systemic inflammatory diseases [20,21,22,23] and its association with reduced *Bifidobacterium* abundance [8,14,16], we aimed to assess the effectiveness of probiotic *B. longum* subsp. *longum* in reducing cigarette smoke-induced inflammation in mice. We utilized two strains of *B. longum* subsp. *longum* which differed in their acetate production capacity and facilitated the investigation of the role of acetate in mediating the effects. Airway and parenchymal inflammation were assessed after cigarette smoke exposure. The gene expression of inflammatory cytokines and adhesion factors was also assessed, and the SCFA abundance was quantified in caecum contents. This demonstrated that both the strains of *B. longum* subsp. *longum* alleviated cigarette smoke-induced inflammation and the expression of cytokines and adhesion factors, which is associated with protection against cigarette smoke-induced depletion in caecum butyrate levels.

## 2. Results

Female C57BL/6 mice were administered either vehicle (PBS + 0.05% L-cysteine) or *B. longum* subsp. *longum* by gavage three times per week and exposed to cigarette smoke or normal air for 8 weeks before assessments of inflammation, cytokine and adhesion factor gene expression, and cecal SCFA abundance were performed. Two strains of *B. longum* subsp. *longum* were used*:* a wild-type strain (WT; NCC2705) capable of producing acetate and a genetically modified strain (MUT; NCC9036) which has an impaired ability to produce acetate [5].

### 2.1. B. longum subsp. longum Reduced Cigarette Smoke-Induced Airway and Parenchymal Inflammation

Cigarette smoke exposure induced airway inflammation in vehicle-treated mice, with increased total leukocytes, neutrophils, macrophages and lymphocytes in bronchoalveolar lavage fluid (BALF, Figure 1A–D). In the mice administered with either strain of *B. longum* subsp. *longum*, the cigarette smoke exposure also increased the amount of total leukocytes, neutrophils and macrophages (*p* = 0.06; WT), but the magnitude of airway inflammation was significantly lower in the mice receiving the *B. longum* subsp. *longum* MUT strain. The parenchymal immune cells were enumerated in hematoxylin/eosin-stained histopathology sections, and this demonstrated that both strains of *B. longum* subsp. *longum* significantly attenuated cigarette smoke-induced parenchymal inflammation. Thus, the mutant strain of *B. longum* subsp. *longum*, with a lower acetate production potential, attenuated both airway and parenchymal inflammation, whilst wild-type *B. longum* subsp. *longum* attenuated parenchymal inflammation only.

### 2.2. B. longum subsp. longum Reduced Cigarette Smoke-Induced Cytokine and Adhesion Factor Expression

The gene expression of cytokines and adhesion factors was assessed in the whole lung tissue by qPCR. The cigarette smoke exposure increased the mRNA expression of the cytokines tumor necrosis factor-α (*Tnfa*), chemokine (C-C motif) ligand (*Ccl*)8, chemokine (C-X-C motif) ligand 2 (*Cxcl2*), and *Ccl22 (*Figure 2A–D). WT *B. longum* subsp. *longum* significantly attenuated *Tnfa* and *Ccl8* expression (Figure 2A,B), and MUT *B. longum* subsp. *longum* attenuated *Tnfa*, *Ccl8,* and *Cxcl2* expression (Figure 2A–C). However, *Ccl22* expression was not reduced by either strain of *B. longum* subsp. *longum* (Figure 2D). The cigarette smoke also induced increases in the expression of adhesion factors such as vascular cell adhesion molecule-1 (*Vcam1*) and intercellular adhesion molecule-1 (*Icam1*; Figure 2E,F). These increases were alleviated by both the WT and MUT strains of *B. longum* subsp. *longum*, demonstrating that their anti-inflammatory impacts are likely associated with the suppression of cytokine and adhesion factor expression.

### 2.3. B. longum subsp. longum Prevented Cigarette Smoke-Induced Butyrate Depletion

The production of acetate is severely impaired in the MUT *B. longum* subsp. *longum* strain, which appeared to have greater anti-inflammatory impacts compared to those of the WT strain. The cigarette smoke exposure increased the total SCFA abundance in the mice treated with WT, but not in those treated with MUT *B. longum* subsp. *longum* (Figure 3A). This effect was primarily driven by acetate, the amount of which was significantly increased by the cigarette smoke exposure in both the vehicle and WT *B. longum* subsp. *longum*-treated mice, but not in the MUT *B. longum* subsp. *longum*-treated mice (Figure 3B). Propionate abundance was highly variable and not significantly altered by either the cigarette smoke exposure or treatment (Figure 3C). However, cigarette smoke exposure reduced caecum butyrate (Figure 3D), which was partially alleviated by the treatment with WT *B. longum* subsp. *longum*. This also corresponded to an increase (*p* = 0.0503) in butyrate in the mice treated with WT *B. longum* subsp. *longum* and exposed only to normal air. Interestingly, MUT *B. longum* subsp. *longum* completely reversed the cigarette smoke-induced depletion of butyrate, to the extent that the cigarette smoke increased the butyrate levels in the MUT *B. longum* subsp. *longum*-treated mice to be greater than those of the air-exposed controls. Overall, the anti-inflammatory effects of *B. longum* subsp. *longum* were associated with protection against the cigarette smoke-induced depletion of butyrate, which was most pronounced in the mice treated with MUT *B. longum* subsp. *longum*.

## 3. Discussion

Overall, these results demonstrate that probiotic *B. longum* subsp. *longum* alleviates cigarette smoke-induced lung inflammation in mice, as evidenced by the reduced number of BALF and parenchymal immune cells. While further research is required in specific disease contexts, these findings indicate the use of *B. longum* subsp. *longum* or other probiotics as potential treatments to reduce the risk of developing chronic inflammatory diseases of the lungs. Smokers frequently struggle to stop smoking behavior, and even if smoking cessation is successful, chronic inflammation and microbial dysbiosis persist afterward, which are key drivers of disease [24,25]. Billions of people worldwide are exposed to chronic air pollution, which can exert similar effects [22,26]. Interventions to rectify these chronic mechanisms of pathogenesis, including probiotics, could alleviate this disease burden.

TNFα, which is primarily produced by macrophages, drives numerous inflammatory responses including the upregulation of *Icam1* and *Vcam1* [27,28]. It has been implicated in cigarette smoke-induced airway remodeling, emphysema, the epithelial–mesenchymal transition, and lung cancer [29,30]. *Ccl8* is a monocyte chemoattractant [31], whilst CXCL2 acts largely as a neutrophil chemoattractant [32] and promotes neutrophil adhesion for migration to the sites of inflammation [33]. The production of inflammatory cytokines such as *Tnfa*, *Ccl8,* and *Cxcl2* is a normal acute response to cigarette smoke exposure, but chronic exposure leads to persistent inflammation that drives chronic respiratory disease [21,29,32,34,35], suggesting that there are benefits to alleviation by *B. longum* subsp. *longum*. However, the protective effects of *B. longum* subsp. *longum* were not universal, as neither strain alleviated the cigarette smoke-induced expression of *Ccl22*.

The adhesion molecules *Icam1* and *Vcam1* contribute to the adhesion and migration of immune cells from the circulation into lung tissue, and their reduced expression may contribute to the anti-inflammatory effects of *B. longum* subsp. *longum*. In addition, smokers are at increased risk of respiratory infections [36,37], and *Icam1* is an adhesion target for bacterial and viral pathogens such as *Haemophilus influenzae* [38] and rhinovirus [39]. *B. longum* subsp. *longum* has been associated with protection against both bacterial [40] and viral infections in mice [3,41], and the use of *Bifidobacterium* probiotics reduces the incidence of respiratory infections in humans [42,43,44]. Similarly, *Icam1* has also been implicated in tumor metastasis in the lung [45], and both *Icam1* and *Vcam1* contribute to atherosclerosis [46], which are diseases associated with cigarette smoking [34]. Although these findings of a reduced mRNA expression of cytokines and adhesion factors indicates a potential mechanism by which *B. longum* subsp. *longum* alleviates cigarette smoke-induced inflammation, we did not investigate the changes in the proteins. The validation of changes in the protein abundance should be further investigated.

Interestingly, the MUT *B. longum* subsp. *longum* strain, which has a severely impaired ability to produce acetate, successfully alleviated all of the measures of lung inflammation, and unlike the WT strain, it even reduced the BALF immune cell influx. Thus, the mechanism of protection was independent of acetate production. Indeed, in contrast to the findings in humans [15] and rats [16], the total SCFAs were not decreased by the cigarette smoke exposure, but they were increased by it, which was driven largely by the increased acetate levels in the vehicle and WT *B. longum* subsp. *longum-*treated mice. Other models in mice have identified no impact of cigarette smoke on the fecal SCFA levels, albeit with the concurrent administration of poly I:C, which suggests that cigarette smoke-induced changes in SCFA abundance are dependent on the experimental conditions [47].

The presence or absence of particular bacteria, such as the nicotine-degrading *Bacteroides xylanisolvens* can substantially alter the responsiveness of the microbiome to exogenous challenges such as cigarette smoke [13]. Moreover, host–microbiome interactions are bi-directional, and the characteristics of the host can influence the effects of microbiota and vice versa. For example, heat-inactivated *Bifidobacterium* spp. isolated from allergic infants induced greater pro-inflammatory responses than those did from healthy individuals [48]. Cigarette smoke can directly affect the virulence of bacteria [49], and it alters the growth, metabolism, and exopolysaccharide structure of *B. animalis* [17]. Given that cigarette smoke exposure causes a gastrointestinal pathology [35,50,51,52], there is likely a direct influence of cigarette smoke on local microbiota which is further influenced by host–microbe and microbe–microbe interactions in this complex system. Thus, it is likely that the host species and/or environment-associated differences in the microbiome composition account for the differing effects of cigarette smoke exposure on SCFA abundance between the previous studies and our current findings.

Although there was no cigarette smoke-induced depletion of caecum acetate and *B. longum* subsp. *longum* did not increase the amount of acetate, WT *B. longum* subsp. *longum* partially alleviated the cigarette smoke-induced butyrate depletion and MUT *B. longum* subsp. *longum* increased the amount of butyrate in the cigarette smoke-exposed mice. *B. longum* subsp. *longum* is not a butyrate producer, but it can increase the rate of butyrate production through the cross-feeding of bacteria containing butyryl CoA:acetate CoA-transferase [53]. Thus, *Bifidobacterium* species can co-operate with other members of the microbiome to more efficiently digest complex carbohydrates, facilitating an increased availability of nutrients for butyrogenesis by other commensal microorganisms [54].

Finally, while increased butyrate abundance may contribute to the anti-inflammatory effects of *B. longum* subsp. *longum*, *Bifidobacterium* species can directly interact with the hosts’ immunity independent of the SCFAs. *B. breve* reduced the inflammatory responses in macrophages exposed to cigarette smoke extract in vitro [6], and mice treated intranasally with exopolysaccharide from *B. longum* subsp. *longum* stimulated TLR2 to promote allergic tolerance through IL-10 and an increased M1/M2 macrophage ratio [7]. Such anti-inflammatory effects help to maintain homeostatic immune tolerance to commensal gut microbiota [55], and they may directly influence the immune cells in the lung if ligands enter the circulation through cigarette smoke-induced “leaky” epithelial barriers [50]. Interestingly, cigarette smoke alters the structure of the TLR2-agonist exopolysaccharide [17], and TLR2 protects against cigarette smoke-induced lung pathology [56], but whether this is associated with stimulation by the gut microbiota is unclear.

Overall, this study demonstrates that the probiotic administration of *B. longum* subsp. *longum,* irrespective of their acetate-producing capacity, alleviated cigarette smoke-induced inflammation and the depletion of cecal butyrate levels. Further research in specific disease contexts will aid in determining whether this is a viable intervention.

## 4. Materials and Methods

### 4.1. Mice, Cigarette Smoke Exposure, and Probiotic Treatment

Female C57BL/6 mice (6–8 weeks old) were obtained from Australian Bioresources (Moss Vale, Australia). The mice were exposed to normal air or the smoke of twelve 3R4F reference cigarettes (University of Kentucky, Lexington, KY, USA) in a custom-designed, purpose-built, nose-only inhalation apparatus (CH Technologies, Westwood, NJ, USA)twice per day, 5 days per week, for 8 weeks, as previously described [24,51,52,57,58,59,60,61,62,63,64]. The mice were treated with 3 × 10^8^ colony forming units (cfu) of *B. longum* subsp. *longum* by intragastric gavage, with non-treated mice receiving a vehicle (PBS + 0.05% L-cysteine, 150 μL). Two strains were utilized: *B. longum* subsp. *longum* NCC2705 (WT), or the genetically modified strain *B. longum* subsp. *longum* NCC9036 (MUT), where the sugar ABC transporter solute-binding protein BL0033 was disrupted by insertional mutagenesis, causing a significantly reduced capacity to produce acetate [5]. All of the experiments were approved by the University of Newcastle Animal Care and Ethics Committee (A-2013–303).

### 4.2. Airway Inflammation

Airway inflammation was quantified by the total and differential enumeration of inflammatory cells in BALF [24,56,57,58]. Briefly, two 0.4 mL washes with PBS of the left lung were performed, the red blood cells were removed by lysis, and the total inflammatory cells counted, cytospun, air dried and stained with May–Grunwald–Giemsa stain for differential counts.

### 4.3. Parenchymal Inflammation

For the histological analysis, the lung tissue was perfused with saline administered via a cardiac puncture, inflated (500 µL), and fixed in formalin prior to mounting, sectioning, and staining. Parenchymal inflammation was assessed from hematoxylin and eosin (H&E)-stained lung sections by counting the number of inflammatory cells in 10 randomized fields of view at 100× magnification [24,56,58,65].

### 4.4. RNA Extraction, Reverse Transcription, and qPCR

Freshly excised lung and colon tissues were snap frozen for subsequent storage at 80 °C. The RNA was extracted using standard protocols [50,56,58,59,65]. Briefly, the tissue was thawed and homogenized (1 mL of TRIzol, 4 °C; ThermoFisher Scientific, Scoresby, Australia) using a tissue tearor homogenizer (Biospec Products, Bartlesville, OK, USA). The DNA was precipitated by chloroform addition, which was followed by centrifugation (12,000× *g*, 15 min, 4 °C), and the RNA-containing aqueous phase was collected. The RNA was precipitated by the addition of isopropyl alcohol and pelleted (12,000× *g*, 10 min, 4 °C) prior to 2 washes with 75% ethanol. The RNA pellets were dissolved in nuclease-free water, and the RNA purity and quantity were assessed by absorbance at 260 and 280 nm using a Nanodrop spectrophotometer. For the cDNA synthesis, the RNA (1000 ng) was pre-incubated with 1 unit of DNAse I (Sigma-Aldrich, Macquarie Park, Aus; 15 min, room temperature). The samples were heated (10 min, 65 °C) prior to the addition of random hexamer primers (50 ng, Meridian Bioscience, Memphis, TN, USA) and dNTPs (10 mM, Meridian Bioscience, Memphis, TN, USA), and it was further heated (5 min, 65 °C). Dithiothreitol (10 mM) and Bioscript reverse transcriptase (200 units in reaction buffer, Meridian Bioscience, Memphis, TN, USA) were added, and reverse transcription was performed (10 min, 25 °C; 50 min, 42 °C; 15 min, 70 °C). The qPCR analysis was performed in 384-well plates with primers for specific transcripts (Table 1) and SYBR-green-based detection using a Viia 7 Real Time PCR system (ThermoFisher Scientific, Scoresby, Australia). Data are expressed as the relative abundance compared to hypoxanthine-guanine phosphoribosyltransferase (*Hprt*).

### 4.5. SCFA Quantification

The SCFA quantification was performed using established methods [66,67]. Briefly, caecum contents were mixed thoroughly with ultrapure water (2000 μL) by mechanically disrupting the contents using a pipette tip and vortexing them. The extracts were passed through 0.22 μm filters, and an aliquot of sample (100 μL) mixed with 10% formic acid (11 μL) and injected into a gas chromatograph with a polar capillary column (DB-FFAP; Agilent, Santa Clara, CA, USA) at 140 °C and a flame ionization detector at 250 °C. A standard calibration curve was used to quantify the SCFAs by the peak area.

### 4.6. Statistical Analysis

The statistical analysis was performed using GraphPad Prism v9.0 (San Diego, CA, USA), including the identification of the outliers using Grubbs test. The data were analyzed by one-way ANOVA with Holm-Šídák’s post hoc test.

## Figures and Tables

**Figure 1 ijms-24-00252-f001:**
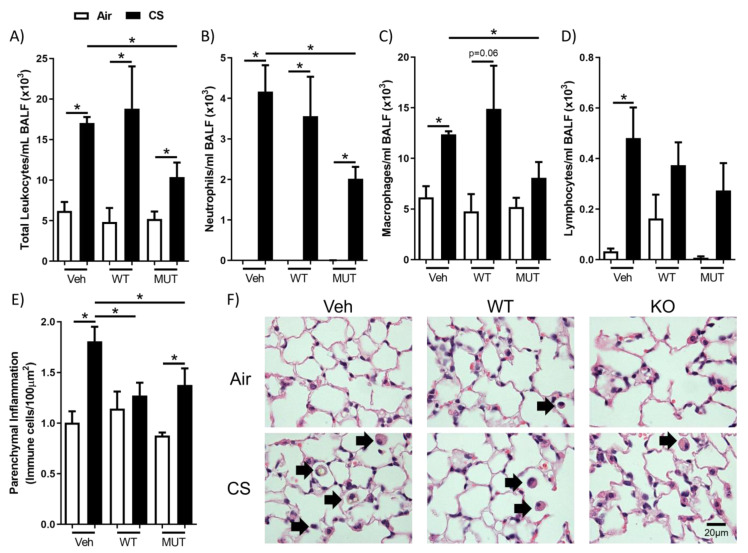
Probiotic *Bifidobacterium longum* subsp. *longum* reduced cigarette smoke-induced lung inflammation. Mice received wild-type (WT) or mutant (MUT) *B. longum* subsp. *longum* or vehicle (Veh) by oral gavage and were exposed to cigarette smoke (CS; black) or normal air (Air; white) for 8 weeks. (**A**) Total leukocytes, (**B**) neutrophils, (**C**) macrophages, and (**D**) lymphocytes were quantified in bronchoalveolar lavage fluid (BALF). (**E**) Immune cells in lung parenchyma were quantified in hematoxylin/eosin-stained lung sections. (**F**) Representative histology images. Black arrows indicate immune cells. Data are presented as mean ± SEM. N = 5–6. * = *p* < 0.05.

**Figure 2 ijms-24-00252-f002:**
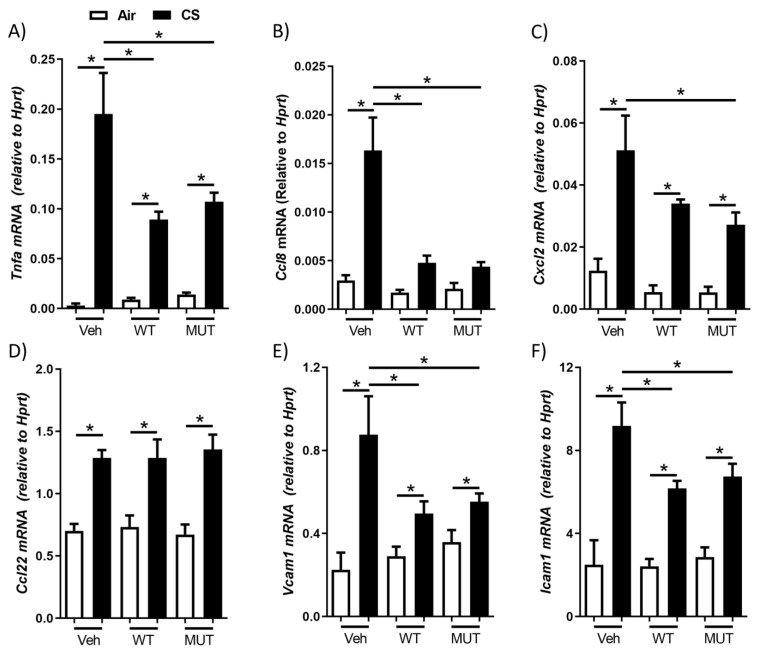
Probiotic *Bifidobacterium longum* subsp. *longum* suppressed cigarette smoke-induced cytokine and adhesion factor gene expression. Mice were treated with wild-type (WT) or mutant (MUT) *B. longum* subsp. *longum* or vehicle (Veh) by oral gavage and were exposed to cigarette smoke (CS: black) or normal air (Air: white) for 8 weeks. (**A**) *Tnfa*, (**B**) *Ccl8, (***C***) Cxcl2*, (**D**) Ccl22, (**E**) *Vcam1,* and (**F**) *Icam1* gene expression relative to *Hprt* was assessed by qPCR in lung tissues. Data are presented as mean ± SEM. N = 5–6. * = *p* < 0.05.

**Figure 3 ijms-24-00252-f003:**
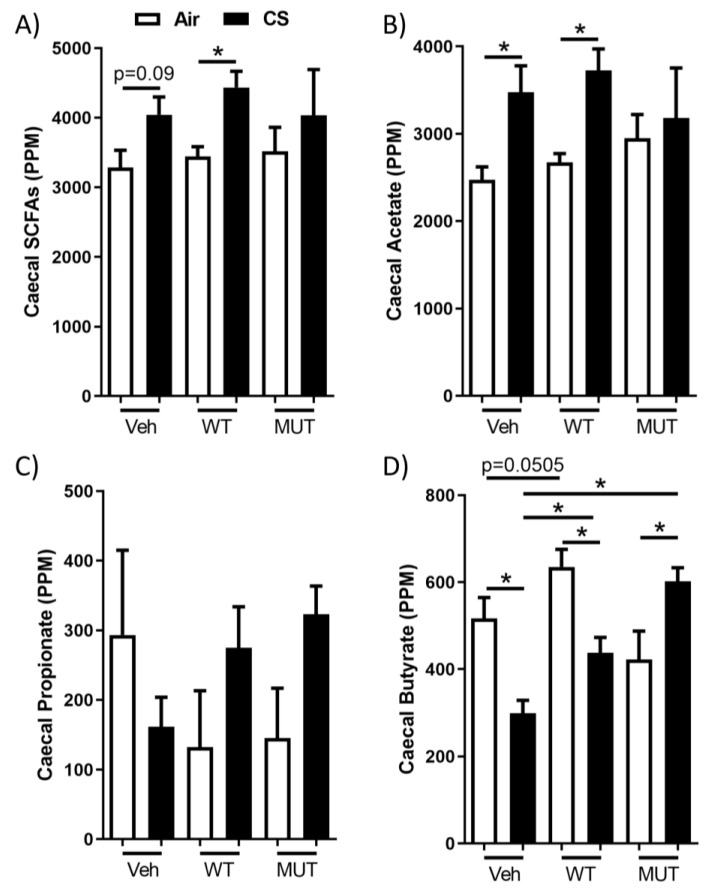
Probiotic *Bifidobacterium longum* subsp. *longum* protected against cigarette smoke-induced decreases in cecal butyrate. Mice treated with wild-type (WT) or mutant (MUTMUT) *B. longum* subsp. *longum* or vehicle (Veh) by oral gavage and were exposed to cigarette smoke (CS: black) or normal air (Air: white) for 8 weeks. (**A**) Total levels of short chain fatty acids (SCFAs), (**B**) acetate, (**C**) propionate, and (**D**) butyrate were quantified in cecal contents by gas chromatography. Data are presented as mean ± SEM. N = 5–6. * = *p* < 0.05.

**Table 1 ijms-24-00252-t001:** List of primers for qPCR.

Target	Forward Sequence (5′–3′)	Reverse Sequence (5′–3′)
*Hprt*	AGGCCAGACTTTGTTGGATTTGAA	CAACTTGCGCTCATCTTAGGCTTT
*Tnfa*	TCTGTCTACTGAACTTCGGGGTGA	TTGTCTTTGAGATCCATGCCGTT
*Ccl8*	GCAGCAGGTGACTGGAGCCT	GCCTGCTGCTCATAGCTGTCCC
*Cxcl2*	TGCTGCTGGCCACCAACCAC	AGTGTGACGCCCCCAGGACC
*Ccl22*	TGGCTACCCTGCGTCGTGTCCCA	CGTGATGGCAGAGGGTGACGG
*Vcam1*	CCCACCATTGAAGATACCGGGA	TAGTATAGGAGAGGGGCTGACC
*Icam1*	GCCTTGGTAGAGGTGACTGAG	GACCGGAGCTGAAAAGTTGTA

## Data Availability

The data in this study are available upon request from the corresponding author.

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
