# Peer review of "Probiotic Bifidobacterium longum subsp. longum Protects against Cigarette Smoke-Induced Inflammation in Mice"

_ijms, 2022, doi:10.3390/ijms24010252_

Round 1

Reviewer 1 Report

Summary of the paper

A manuscript by Budden et al. explores whether Bifidobacterium longum supplementation alleviates smoke-induced inflammation in mice. The authors demonstrate that supplementation of B. longum in mice is associated with reduced lung inflammation, inflammatory cytokine expression, and adhesion factor expression, and alleviated cigarette smoke-induced depletion in caecum butyrate. Overall, this work will be of interest to researchers and health professionals working in the field, given that the gut-lung axis is not often described in the literature. I only have some minor suggestions.

Minor remarks

·      Lines 17, 29, 32, 45: "Bifidobacteria". The correct name of the genus is Bifidobacterium

·      Line 49 and forward: please specify the subspecies of Bifidobacterium longum (it contains at least three different ones). I assume the authors meant Bifidobacterium longum subsp. longum

·      Figure 1F: indicate in the legend what is pointed by black arrows.

·      111-113: "Cigarette smoke exposure induced a trend towards increased total SCFA abundance (p=0.09) in vehicle-treated mice..."
Please rephrase or omit. Emphasizing observations that "trend towards statistical significance" is a misleading practice because it implies that increasing the sample size would result in better p-values, which is often not true (see https://doi.org/10.1136/bmj.g2215)

·      205-206: "Overall, this study demonstrates that both acetate- and non-acetate-producing B. longum"... Please rephrase. The BL0033 knockout strain is still capable of producing acetate (see the original Fukuda et al. paper)

·      Line 217-218: the strain did not "lack" the BL0033 gene; it was disrupted by insertional mutagenesis

Reviewer 2 Report

The manuscript “Probiotic Bifidobacterium longum protects against cigarette smoke-induced inflammation in mice” by Budden et al. They have reported the anti-inflammatory effects Bifidobacterium longum against cigarette smoke using a mice model. The authors have done various parameters to prove their hypothesis. The manuscript is written in standard English with several grammatical and typographical errors. After thoroughly reviewing I feel the manuscript needs revision.

Comments:

1.     In the abstract section, I will suggest rewriting the conclusion in a better way.

2.     I will suggest adding briefly the reason behind the selection of acetate forming WT and KO B. longum in the introduction section.

3.     I will suggest that authors can see a few more inflammatory (IL6) and anti-inflammatory cytokines IL10.

4.     Authors have performed only gene expression to prove their hypothesis…I will suggest that authors can add western blot or ELISA that can add values to the data and manuscript.

5.     I will suggest adding a graphical abstract that can be the point of interaction for the readers.

Round 2

Reviewer 2 Report

Authors have responded my comments and suggestions positively.